# Increased Material Density within a New Biomechanism

**Carlos Aurelio Andreucci** [1], **Elza M. M. Fonseca** [2,*] and **Renato N. Jorge** [1]

1 Mechanical Engineering Department, Faculty of Engineering, University of Porto, Rua Dr. Roberto Frias, 712, 4200-465 Porto, Portugal

2 Mechanical Engineering Department, School of Engineering, Polytechnic Institute of Porto, R. Dr. António Bernardino de Almeida 431, 4200-072 Porto, Portugal

* Correspondence: elz@isep.ipp.pt

**Abstract:** A new mechanism, applied in this study as a biomechanical device, known as a Bioactive Kinetic Screw (BKS) for bone implants is described. The BKS was designed as a bone implant, in which the bone particles, blood, cells, and protein molecules removed during bone drilling are used as a homogeneous autogenous transplant at the same implant site, aiming to optimize the healing process and simplify the surgical procedure. In this work, the amount of bone that will be compacted inside and around the new biomechanism was studied, based on the density of the bone applied. This study allows us to analyze the average bone density in humans (1.85 mg/mm$^3$ or 1850 µg/mm$^3$) with four different synthetic bone densities (Sawbones PCF 10, 20, 30 and 40). The results show that across all four different synthetic bones densities, the bone within the new model is 3.45 times denser. After a pilot drill (with 10 mm length and 1.8 mm diameter), in cases where a guide hole is required, the increase in ratio is equal to 2.7 times inside and around the new biomechanism. The in vitro test validated the mathematical results, describing that in two different materials, the same compact factor of 3.45 was determined with the new biomechanical device. It was possible to describe that BKS can become a powerful tool in the diagnosis and treatment of natural bone conditions and any type of disease.

**Keywords:** Bioactive Kinetic Screw; bone mass density; osteopenia; osteoporosis; DEXA

## 1. Introduction

Bone tissue is frequently remodeled through bone resorption by osteoclasts and bone formation by osteoblasts, having osteocytes as mechano-sensors and organizers of the bone remodeling process. This mechanism is regulated by growth factors, cytokines, and systemic factors that achieve bone homeostasis [1]. Any dysregulation in this process can cause osteoporosis.

Osteoporosis is a metabolic pathology defined by bone mass reduction related with reduced bone strength and heightened skeletal fragility with bone tissue microarchitectural degradation. Osteoporosis is a frequent cause of bone fractures in the elderly [2]. Dual-energy-x-ray absorptiometry (DEXA) can grant an accurate diagnosis of osteoporosis and assessment of fracture risk. The World Health Organization has settled DEXA as the standard technique for measuring bone mineral density (BMD) [3].

Bone Screws in orthopedic surgery convert torsional forces into compression. The primary functional objective in the design of a screw is to dissipate and distribute the mechanical load. Thread design should maximize initial contact, enhance surface area, dissipate, and distribute stresses at the screw–bone interface and increase the pullout strength. Screws can be used for attachment of implants to bone, bone-to-bone fixation, or for soft tissue fixation or anchorage [4]. Bone volume and bone quality properties are fundamental prerequisites to ensure optimal mechanical stability of the implants and further osseointegration [5].

In the new biomechanism (BKS), Figure 1, the hard tissue is removed during bone drilling and screwed at the same time, due to the new characteristics, which uses the chips (bone particles, blood, cells, and protein molecules) to fill the hole passing through the drill grooves in the new screw. The screw was created based on the concepts of using drills for bone drilling and screws for fixation and implants [4]. Traditional drill flutes remove bone particulate after perforation. In the BKS model, a mechanical limit was created for these flutes through a hole joining the cut grooves, avoiding the removal of all particulate bones from the perforation. This bone material is compacted in and through the screw, increasing the bone contact surface and creating a graft bone bridge of particulate material through the screw, according to the authors project [6].

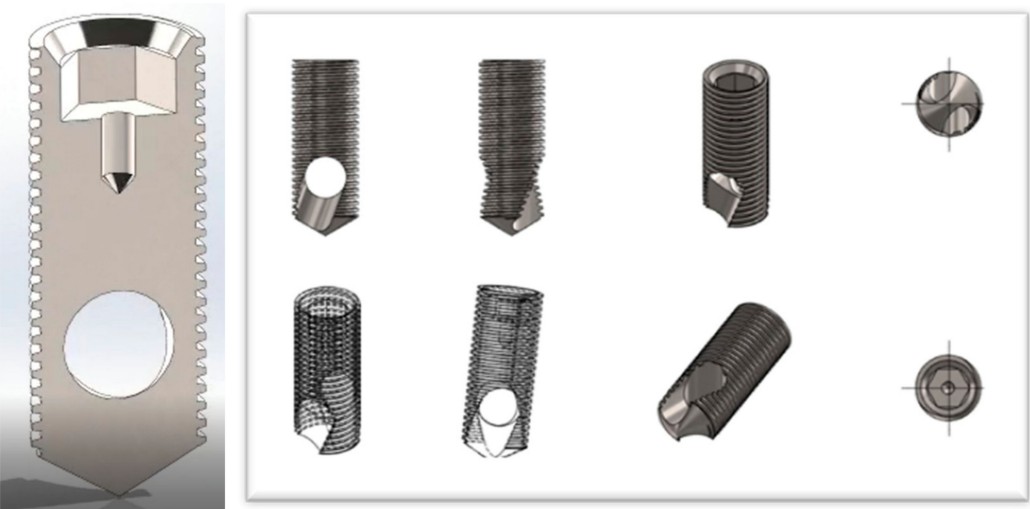

**Figure 1.** BKS of 4 mm width and 10 mm length, according to the authors project [6].

Developing a biomechanism for bone fixation that involves biological and mechanical characteristics that are desirable in biomechanical rehabilitation in a safe and practical way, are the founding principles of all the specialties involved in these studies, from research with molecules and proteins, to genetic therapies for the biomechanical functional restoration of limbs and organs [5,6].

In this ongoing research, as in the previous work from the authors [6], the main objective is to simulate and test in vitro and in vivo the new biomechanical function described by the new BKS model, applying the concepts of a modified drilling, and screwing in the same device. The new biomechanism will be mathematically described, and its fundamental study is to help and guide practical decisions and have a protocol to be validated. The new screw refers to a line of research initiated through a new biomechanical model for "Improvement in screw of fixations and bone implant" [6] developed to provide the fixation of osseointegrated screws, to facilitate and simplify the surgical protocol of insertion and fixation, obtaining primary stability and consequent secondary stability, through greater contact of the recipient bone with the surface of the new fixation screw and through it. The filling through the screw hole, Figure 1, with the bone of the bed site to be drilled, used in the screw of the drills of the cross flutes system, limited by a through hole, allows the use of the bone present in the flutes in the moment of screwing-drilling; the chips (particulate bones) flow as a homogenous autogenous transplant that will be inserted and compacted in the through hole. With this, physiological bone healing stimulus are obtained (particulate bone, cells, and molecules through the screw or implant) [7–9], optimizing natural bone healing by using the organic bone tissue repair to aid in integration, stability, and longevity of fixations, with direct bone contact through the implants, making it possible to be a unique and functional object. This could become a biological osseointegration.

## 2. Materials and Methods

In this work, the mean bone density in humans was analyzed (1.85 mg/mm$^3$ or 1850 µg/mm$^3$) [10] with four different synthetic bone densities, such as closed cell polyurethane (PCF) from Sawbones with the references PCF 10, 20, 30, and 40, respectively, simulating natural bone density in four macroscopic classes (D1, D2, D3, and D4) [11].

PCF foam is used as an alternative test medium for human cancellous bone. It does not replicate the structure of human bone; however, it provides properties consistent in the range of human cancellous bone. PCF is mostly used to test screw pullout, insertion, and stripping torque at different densities.

To measure the volume of the internal space of the new biomechanical model in SolidWorks software, the volume of the complete screw design was used, as shown in Figure 2, and the volume with the new biomechanism was subtracted, Figure 3. According to the procedure, the entire new internal volume of the biomechanical device is equivalent to 28.10 mm$^3$. The difference between bone mass (m) and bone density (D) is that the first refers to the amount of bone tissue in the skeleton, and bone density refers to the mineral mass per unit volume of bones [12–14]. Density (D) can be calculated using Equation (1) related to volume (V). Density is commonly expressed in units of grams per cubic centimeter.

$$D = m/V \tag{1}$$

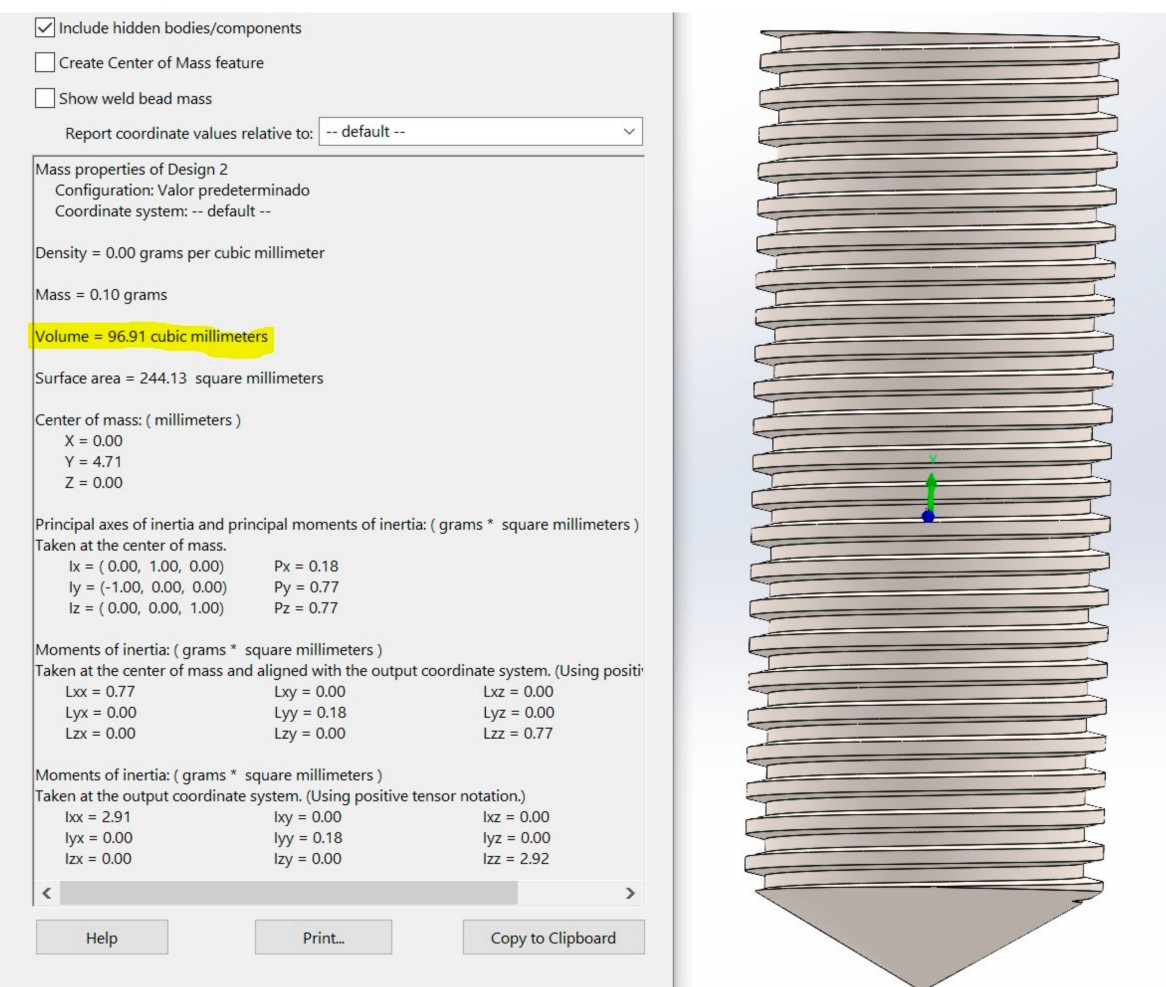

**Figure 2.** Complete model with a volume equal to 96.91 mm$^3$.

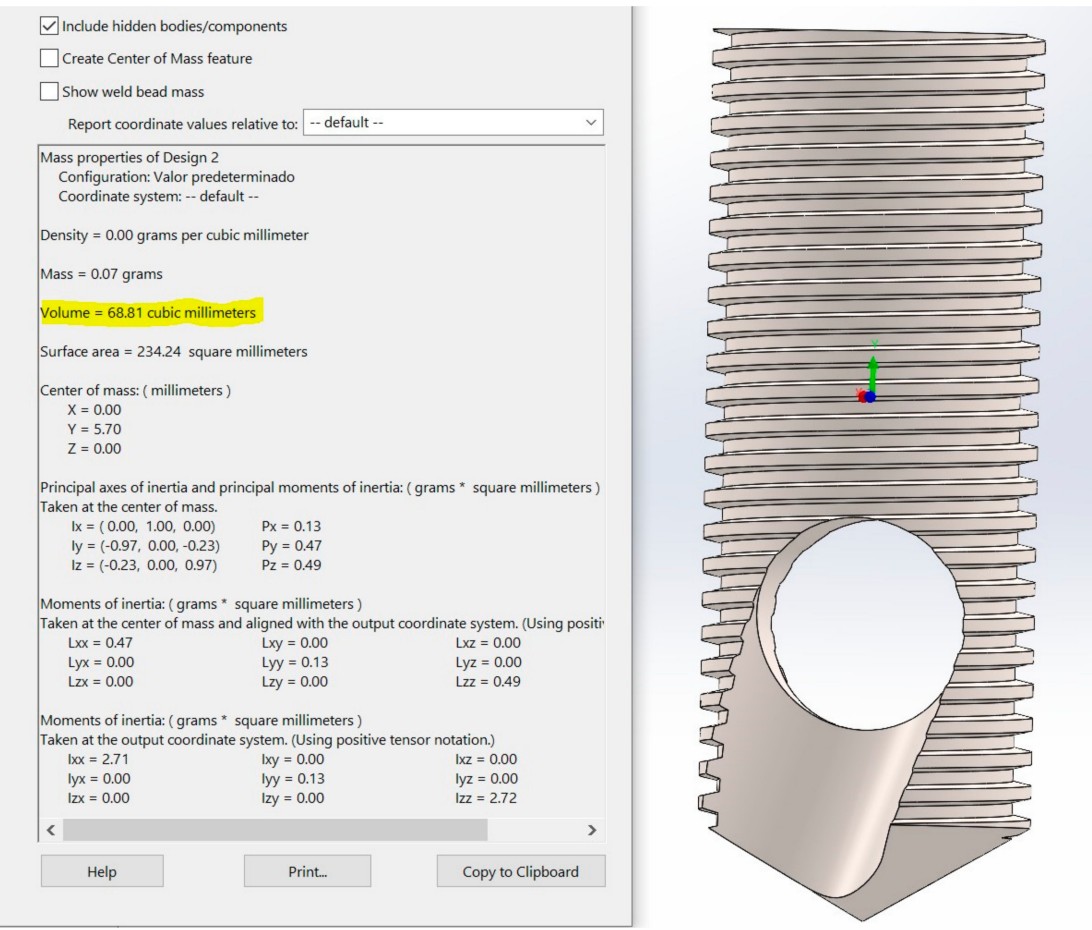

**Figure 3.** Final model with new biomechanism with a volume equal to 68.81 mm$^3$.

To validate the mathematically proved concept, the new BKS model was 3D printed in Polylactic Acid (PLA), as shown in Figure 4, to be experimentally tested in Aluminum Phyllosilicates (clay) and Kinetic Sand. The BKS device was weighted before being manually screwed and unscrewed in the materials with a regular hexagonal spanner wrench. After collecting and compacting the material inside the BKS, another weight measurement was taken with all the material compacted. The difference between the weights before and after was determined by a Taylor Precision Compact Digital scale, using the calibration button before all weight measurements.

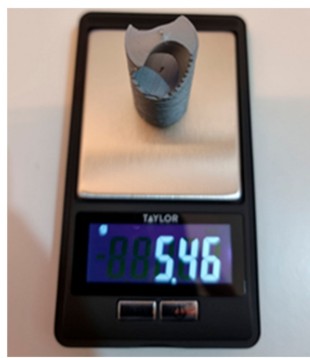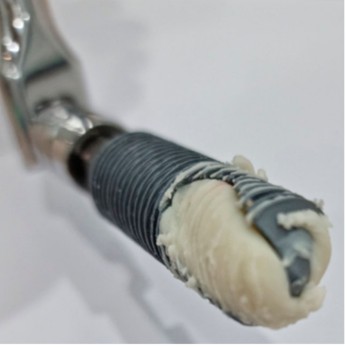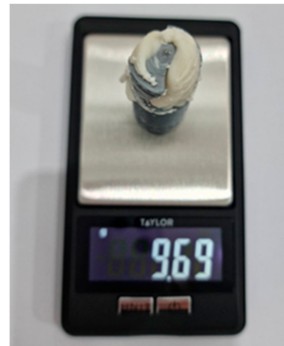

**Figure 4.** BKS device 3D printed and Taylor Precision Compact Digital scale before and after compacting Aluminum Phyllosilicates (clay).

Six measurements were performed in each material, obtaining an average of values, which was compared with the normal density of the materials itself and with the compaction factor described in the mathematical proof, validating it.

## 3. Results

### 3.1. Average Density in Human Bone

The volume inside the new biomechanism, measuring 4 mm in diameter and 10 mm in length, is 28.10 mm$^3$. Thus, with bone density equal to 1850 µg/mm$^3$, the calculated mass is 51,985 µg, approximately 0.05 g.

With bone density equal to 1850 µg/mm$^3$, a total of 51,985 µg is determined inside and around the new biomechanism (28.1 mm$^3$) for natural bone density, without measuring the effect of chip (particulate bone) compacting effect of the new biomechanism after cutting and insertion (total screw volume).

The total volume of the model is 96.91 mm$^3$. The total amount of bone collected to be compacted will be the total volume of the screw multiplied by the density of the bone equal to 1850 µg/mm$^3$. The mass obtained is equal to 179,283.50 µg. This value represents 0.18 g to be compacted into 28.10 mm$^3$. Thus, with the effect of compacting inside the screw, a mass of 179,283.50 µg in the volume 28.10 mm$^3$ was calculated, which allows a density calculation equal to 6380.19 µg/mm$^3$ or 6.38 mg/mm$^3$.

The density inside the screw after compacting bone will be 6.38 mg/mm$^3$; in the same volume with natural bone density, it was equal to 1.85 mg/mm$^3$. This means a density grow by 3.45 times inside and around the new biomechanical model optimizing the osseointegration is to be tested in vivo [15].

### 3.2. Average Density Using PCF

To test the amount of bone compacted in the BKS system in PCF foam material, in four different densities, the same introduced concept was used. Table 1 represents the chosen PCF densities, according to ASTM D1622 [16], the calculated results for the mass of the total screw drilled (volume of the model is 96.91 mm$^3$) and the density of the total drilled screw compacted inside the BKS screw (in the volume 28.10 mm$^3$).

**Table 1.** Calculated values for chosen PCF densities.

| PCF | PCF Density, g/cm$^3$ [10] | Mass, mg | Compacted Density, mg/mm$^3$ |
|-----|---------------------------|----------|------------------------------|
| 10 | 0.16 | 15.50 | 0.55 |
| 20 | 0.32 | 31.01 | 1.10 |
| 30 | 0.48 | 46.51 | 1.65 |
| 40 | 0.64 | 62.02 | 2.20 |

With the new BKS, a compacting effect was identified, making the synthetic bone inside the screw be 3.45 times denser in all types of densities (10, 20, 30 and 40 PCF) in a volume that represents almost 1/3 (28.10 mm$^3$) of the whole screw (96.91 mm$^3$).

BKS has a compacting factor increasing the density on the average human bone and synthetic bones equal to 3.45 times inside and around the new biomechanical model. It means that no matter the value of the bone density, the new biomechanical device increases 3.45 times the bone density in 29% of the total volume of the BKS in the deepest part of the perforation, usually attached in trabecular bone in surgeries.

### 3.3. In vitro Experiment

The in vitro test validated the mathematical results, describing that in two different materials, the same compact factor of 3.45 was determined inside the new biomechanical device. For this, a BKS model manufactured through 3D printing was experimentally tested in Aluminum Phyllosilicates (clay) and Kinetic Sand, with a mean density of fine silts and clays between 1.1 and 1.6 g/cm$^3$ and Kinetic Sand 0.83 g/cm$^3$ [17,18]. The results, as seen

in Table 2, described the same results determined through the mathematical concept on the calculated mean $(\overline{X})$, each score deviates from the mean by 0.06 points for aluminum phyllosilicates and 0.04 for Kinetic sand. It was possible to also observe the standard deviation (SD) of each one. A small standard deviation was obtained, which indicate the data observed is clustered tightly around the mean.

**Table 2.** Compacting factor analysis in two different materials to test and validate the mathematical proof of concept of the BKS device.

| Material/Average Density(g/cm$^3$) | Volume Inside BKS (g/cm$^3$) | Volume Inside BKS/3.45 (g/cm$^3$) | Mean ($\overline{X}$) and Standard Deviation (SD) Density Inside BKS (g/cm$^3$) |
|---|---|---|---|
| aluminum phyllosilicates/1.10 to 1.60 g/cm$^3$ | 4.23 | 1.22 | |
| | 4.15 | 1.20 | |
| | 4.26 | 1.23 | $\overline{X}$ = 1.23 |
| | 4.19 | 1.21 | SD = 0.02 |
| | 4.27 | 1.24 | |
| | 4.31 | 1.25 | |
| Kinetic sand/0.83 g/cm$^3$ | 2.89 | 0.83 | |
| | 2.90 | 0.84 | |
| | 2.82 | 0.82 | $\overline{X}$ = 0.84 |
| | 2.88 | 0.83 | SD = 0.01 |
| | 2.96 | 0.86 | |
| | 2.90 | 0.84 | |

### 3.4. Using a Pilot Drill in Cases a Guide Hole Is Needed

In the case of using a standard pilot drill with a length of 10 mm and a diameter of 1.8 mm to make a perforation to guide the insertion of the BKS screw, a volume of 20.10 mm$^3$ was removed from the bone bed site. If the volume of the drill (20.10 mm$^3$) is subtracted from the total volume of the screw (96.91 mm$^3$), it will result in 76.81 mm$^3$ as the total volume of bone to be compacted.

Table 3 shows the results of the mass of the total screw drilled at different bone densities after a pilot drilling (volume of 20.10 mm$^3$) and the density of the bone compacted inside the BKS screw at different synthetic bone densities, after a pilot drill (volume of 20.10 mm$^3$).

**Table 3.** Calculated values for chosen PCF densities after a pilot drilling.

| PCF | PCF Density, g/cm$^3$ [10] | Mass, mg | Compacted Density, mg/mm$^3$ |
|---|---|---|---|
| 10 | 0.16 | 12.28 | 0.43 |
| 20 | 0.32 | 24.58 | 0.87 |
| 30 | 0.48 | 36.86 | 1.31 |
| 40 | 0.64 | 49.15 | 1.74 |

BKS has a compacting factor that increases the density, on the average, human bone, and synthetic bones after a pilot drilling (10 mm in length and 1.8 mm in diameter) equal to 2.7 times inside and around the new biomechanical design.

## 4. Discussion

Dual-energy x-ray absorptiometry (DEXA) is the standard test for measuring bone mineral density since its approval by the Food and Drug Administration (FDA) for clinical use in 1988. The Bone Mass Measurements in 1998 solidified its validity considering other diagnostic modalities such as chemical analysis, direct dissection, quantitative ultrasonography, and later, CT/MRI images [19].

DEXA uses the attenuations of low and high-energy photon emissions that are detected above the patient and are combined to create a planar image to assess bone mass per unit volume (g/cm), for example, bone mineral density (BMD) [19,20].

World Health Organization (WHO) in 1994 provided a definition of bone mass loss using a standardized score, called T-scores as: greater than or equal to $-1.0$ is normal; less

than −1.0 to greater than −2.5 considered as osteopenia; less than or equal to −2.5 as osteoporosis and less than or equal to −2.5 plus fragility fracture considered severe osteoporosis.

Correct interpretation of BMD requires attention to detail in anthropometric information, patient positioning, correct scan analysis, BMD pattern of individual vertebrae, and identification of artefacts [21].

With the new biomechanical screw, it is possible to directly access the BMD in cases where more reliable data are needed and when the clinical findings do not match with imaging tests (DEXA). The BKS screw can be used as a bone collector if removed immediately after full bone insertion. The amount of bone inside the new biomechanical screw will be measured and the BMD can be determined, according to the proposed Equation (2) for the ideal condition or Equation (3) when the guided hole is needed. As the bone inside the BKS is exactly 3.45 times denser than the bone of the same bed site, the proposed equation as the follow:

$$BMD = \frac{Bone\ inside\ BKS}{3.45} \tag{2}$$

$$BMD = \frac{(Bone\ inside\ BKS - Bone\ volume\ pilot\ drill)}{3.45} \tag{3}$$

The average bone density in humans is 1.85 mg/mm$^3$ [10]. The density inside the screw after compacting bone will be 6.38 mg/mm$^3$; in the same volume with natural bone density, the value is 1.85 mg/mm$^3$.

For BMD calculation using the BKS screw, the vertebra L1 will be used as an example, as seen in a regular DEXA exam in Figure 5 [20]. The area is 8.92 cm$^2$, the bone mineral content (BMC) is 6.50 g, so the volume is equal to 892 mm$^3$ and the BMC will be 650 mg. Using Equation (1), BMD is equal to 0.244 mg/mm$^3$.

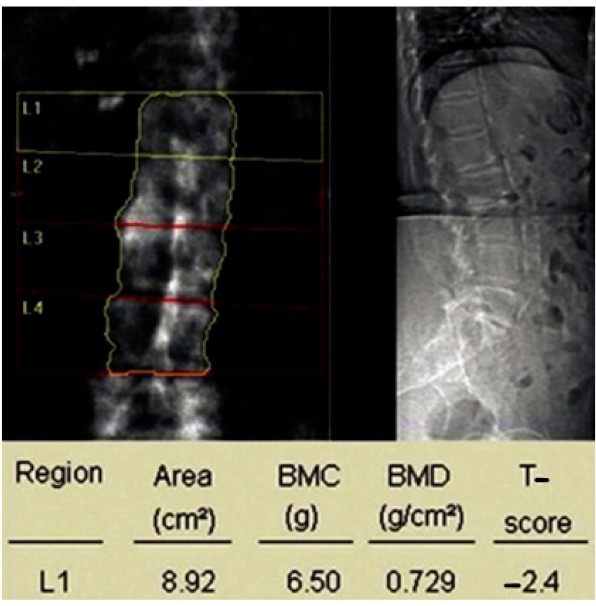

| Region | Area (cm²) | BMC (g) | BMD (g/cm²) | T−score |
|--------|-----------|---------|-------------|---------|
| L1 | 8.92 | 6.50 | 0.729 | −2.4 |

**Figure 5.** Bone mineral density measured and reported on one vertebra L1 [18].

As the bone inside the BKS is exactly 3.45 times denser than the bone in the same bed site, the measure inside the BKS allows a value equal to 0.842 mg/mm$^3$.

In conclusion, Bone Mass Density within the BKS (BKS-BMD) can always accurately define the BMD by dividing its value by 3.45, as per our proposal according to Equation (4).

$$BMD = \frac{BKS - BMD}{3.45} \tag{4}$$

Associating direct bone mass density measure through the new biomechanical screw (BKS) comparing and validating the images results (DEXA) will become a powerful tool in diagnosis and treatment of natural bone conditions and any kind of bone diseases to be tested.

Recent biomechanical studies [22–25] in implant osseointegration showed that the maintenance of bone viability at an osteotomy site is a critical variable for success. Analyzing the consequences of site preparation and the bone loss due the standard protocols, made some authors, including us, to rethink the design of bone-cutting drills for implant site preparation. In our new BKS, the screw is also a drill with a modified compacting factor. Some authors [22] developed a new protocol for drilling applying a new model of drill bit, with the unique design of the cutting flutes, making channels into the osteotomy maintaining on the site autologous bone chips and osseous coagulum that have inherent osteogenic potential. Collectively, these features resulted in robust, new bone formation at rates significantly faster than those observed with conventional drilling protocols.

## 5. Conclusions

A new biomechanism BKS for screws and bone implants developed by the first author was presented using a bone dental implant screw, in which the bone particles, blood, cells, and protein molecules removed during bone drilling are used as a homogeneous autogenous transplant in the same implant site, aiming to obtain primary and secondary bone stability, simplifying the surgical procedure, and improving the healing process. It was observed that at all four different synthetic bones densities the bone inside the new model are 3.45 times denser. After a pilot drill (10 mm in length and 1.8 mm in diameter), in cases where a guide hole is needed, the increased ratio is equal to 2.7 times inside and around the new biomechanical design.

The in vitro test validated the mathematical results, describing that in two different materials, the same compact factor of 3.45 was determined with the new biomechanical device.

With the new biomechanical screw, it is possible to directly access the BMD in cases where more reliable data are needed, during surgery treatments in patients with osteopenia to help confirm the diagnosis and treatment, and when the clinical findings do not match the imaging tests (DEXA), thus becoming a powerful tool in diagnosis and treatment of natural bone conditions and any type of bone disease.

**Author Contributions:** Conceptualization, C.A.A.; methodology, C.A.A.; methodology, C.A.A.; investigation, C.A.A.; writing—original draft, C.A.A.; writing—review and editing, E.M.M.F.; visualization, E.M.M.F.; supervision, R.N.J. All authors have read and agreed to the published version of the manuscript.

**Funding:** This research received no external funding.

**Data Availability Statement:** Not applicable.

**Conflicts of Interest:** The authors declare no conflict of interest.

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
