# Peer review of "Increased Material Density within a New Biomechanism"

_mca, doi:10.3390/mca27060090_

Round 1

Reviewer 1 Report (Previous Reviewer 1)

·         I would recommend authors provide literature on the recent studies on innovative bone screw designs (for example: Yao, Y., Wang, L., Li, J., Tian, S., Zhang, M., & Fan, Y. (2020). A novel auxetic structure based bone screw design: Tensile mechanical characterization and pullout fixation strength evaluation. Materials & design, 188, 108424) to highlight the novelty of the present study.

 ·         The present bone screw design aims to utilize the chips (bone particles, blood, cells, and protein molecules) to fill the hole passing through the drill groove. The use of these chips may lead to infection at the implantation site and earlier failure of the bone screws. I would suggest performing at least some sort of in vitro cytocompatibility/cell viability testing of the screw after the simulated drilling process. For example: Biocompatible filaments may be used for 3D printing the screw and the simulated drilling may be carried out using Simulated body fluid (SBF) inside a pipe or duct or container followed by some sort of cell viability testing. 

Author Response

About the Reviewer’s comments, we answer all your questions in the following lines. The authors would like to thank the Reviewer for his reading work and suggestions for improvements to the manuscript. We have addressed all comments and necessary changes. Changes to the article are identified in blue and green. We hope that the manuscript revisions and accompanying answers are sufficient to make our manuscript suitable for publication.

Review 1

  • I would recommend authors provide literature on the recent studies on innovative bone screw designs (for example: Yao, Y., Wang, L., Li, J., Tian, S., Zhang, M., & Fan, Y. (2020). A novel auxetic structure based bone screw design: Tensile mechanical characterization and pullout fixation strength evaluation. Materials & design, 188, 108424) to highlight the novelty of the present study.

ANSWER 1– The authors thank the reviewer. The Discussion has been rewritten and literature on the recent studies on innovative bone screw designs have been included to support our conclusions and increase the quality of our manuscript:

Chen, C.-H.; Coyac, B.R.; Arioka, M.; Leahy, B.; Tulu, U.S.; Aghvami, M.; Holst, S.; Hoffmann, W.; Quarry, A.; Bahat, O.; Salmon, B.; Brunski, J.B.; Helms, J.A. A Novel Osteotomy Preparation Technique to Preserve Implant Site Viability and Enhance Osteogenesis. J. Clin. Med. 2019, 8, 170. https://doi.org/10.3390/jcm8020170

Yao, Y.; Wang, L.; Li, J.; Tian, S.; Zhang, M.; Fan, Y. A novel auxetic structure based bone screw design: Tensile mechanical characterization and pullout fixation strength evaluation. Mat & design. 2020, 188, 108424

Stahel, P.F.; Alfonso, N.A.; Henderson, C. Introducing the “Bone-Screw-Fastener” for improved screw fixation in orthopedic surgery: a revolutionary paradigm shift?. Patient Saf Surg. 2017, 11, 6. https://doi.org/10.1186/s13037-017-0121-5

Zhang, X.; Tiainen, H.; Haugen, H.J. Comparison of titanium dioxide scaffold with commercial bone graft materials through micro-finite element modelling in flow perfusion. Med Biol Eng Comput. 2019, 57(1):311-324. doi:10.1007/s11517-018-1884-2

  • The present bone screw design aims to utilize the chips (bone particles, blood, cells, and protein molecules) to fill the hole passing through the drill groove. The use of these chips may lead to infection at the implantation site and earlier failure of the bone screws. I would suggest performing at least some sort of in vitro cytocompatibility/cell viability testing of the screw after the simulated drilling process. For example: Biocompatible filaments may be used for 3D printing the screw and the simulated drilling may be carried out using Simulated body fluid (SBF) inside a pipe or duct or container followed by some sort of cell viability testing. 

ANSWER 2– The authors thank the reviewer. The use of the chips (bone particles) is the gold standard for bone grafts in dentistry and dental implants. Even the use of synthetic bones (hydroxyapatite, xenogenic grafts, etc) are well described as a good option for optimizing bone healing. The removal of chips from the dental implant drill bit before the screw implantation, are also very well described protocol in literature (also added the literature into the discussion as the recommendation number 1).

The screw made by titanium is being used since 1979 for dental implants also proved as very safe and this new BKS model will be printed in Commercially Pure Titanium. Nowadays even scaffolds (also cited in the references) with Titanium structures are filled with matrix of bovine bone, mixed with homogenous rich plasma plaquetes with successful outcomes. The innovation here is only using the same time drilling, screwing, and compacting effect inside the screw with particles and increase the density of the material applied. Your great suggestion of in vitro testing will be noted for future research when our goal will be the healing process. We are very thankful for that.

Reviewer 2 Report (New Reviewer)

This paper describes the effects of the material density within a new biomechanism.

The work could be of interest to bioapplication. However, I suggest this review needs revisions before accepting for publication.

1: Line100-101: Is the bone density (D) the same meaning as Density (D)? If not, each expressiion should change. Ex) bone density (BD), Density (D).

2: In this discussion, the detail relationship between results and biomechanism in vivo need to be added. 

3: In this data, statistical analysis should need.

Author Response

About the Reviewer’s comments, we answer all your questions in the following lines. The authors would like to thank the Reviewer for his reading work and suggestions for improvements to the manuscript. We have addressed all comments and necessary changes. Changes to the article are identified in blue and green. We hope that the manuscript revisions and accompanying answers are sufficient to make our manuscript suitable for publication.

Review 2

This paper describes the effects of the material density within a new biomechanism.

The work could be of interest to bioapplication. However, I suggest this review needs revisions before accepting for publication.

1: Line100-101: Is the bone density (D) the same meaning as Density (D)? If not, each expressiion should change. Ex) bone density (BD), Density (D).

ANSWER 1– The authors thank the reviewer. Means the same as Density.

2: In this discussion, the detail relationship between results and biomechanism in vivo need to be added. 

ANSWER 2– The authors thank the reviewer. The detail relationship between results and biomechanism has been included into the Discussion to support and increase the quality of our manuscript with new references.

3: In this data, statistical analysis should need.

ANSWER 1– The authors thank the reviewer. Mean and Standard deviation analysis were applied for the experiment results, table 2.

Round 2

Reviewer 1 Report (Previous Reviewer 1)

Thank you for addressing the comments. I hope to see in vitro and in vivo studies results on BKS later in future studies. 

This manuscript is a resubmission of an earlier submission. The following is a list of the peer review reports and author responses from that submission.

Round 1

Reviewer 1 Report

It is recommended that authors provide a brief introduction to the traditional/existing orthopedic drilling and screws used for bone fixation and implants such that the novelty of the current Bioactive Kinetic Screw (BKS) design is more clearly portraited. 

I would suggest adding a cross-sectional side view of the BKS screw in Figure 1 to show the hole geometry inside the screw. Information on the screw geometry such as overall length, inner and outer diameter, screw pitch, etc. will be helpful.

Tables 1 and 2 include columns 2 and column 4 for densities with the same heading. Clear column headings such as “PCF density and Compacted density” is suggested.

In section 3.1, it is suggested to include the information on screw material and the manufacturing process used for fabricating the BKS screw for pilot drilling.

Equation (2) for bone mineral density (BMD) appears to be valid only for the ideal condition. It is suggested that the author include the equation for cases where the guided hole is needed (based on the pilot drilling) such that a range of BMD can be established instead of a single BMD value.

Figure 4 indicates Z-scores for different Vertebra regions. It is recommended to add information on Z-scores for bone density prior to figure 4.

Author Response

About the Reviewer’s comments, we are answering their concerns in the following lines.

The authors would like again to acknowledge the Reviewer for their work on reading and suggesting improvements to the manuscript. We have addressed all the comments and the changes needed. Changes to the article were identified in blue.

We hope that the revisions in the manuscript and our accompanying answers will be sufficient to make our manuscript suitable for publication in the Mathematical and Computational Applications.

  • It is recommended that authors provide a brief introduction to the traditional/existing orthopedic drilling and screws used for bone fixation and implants such that the novelty of the current Bioactive Kinetic Screw (BKS) design is more clearly portraited.

ANSWER – The authors thank the reviewer. A brief introduction was already added in the article, including the reference 4.

  • I would suggest adding a cross-sectional side view of the BKS screw in Figure 1 to show the hole geometry inside the screw. Information on the screw geometry such as overall length, inner and outer diameter, screw pitch, etc. will be helpful.

ANSWER – A new figure was introduced to clarify the cross-section of the BKS and the dimensions included near to the figure legend.

  • Tables 1 and 2 include columns 2 and column 4 for densities with the same heading. Clear column headings such as “PCF density and Compacted density” is suggested.

ANSWER – Thank you for the suggestion. Already added in the article.

  • In section 3.1, it is suggested to include the information on screw material and the manufacturing process used for fabricating the BKS screw for pilot drilling.

ANSWER – Thank you for the recommendation. We ad “standard” pilot drill, clarifying that there’s no need for any special drill.

  • Equation (2) for bone mineral density (BMD) appears to be valid only for the ideal condition. It is suggested that the author include the equation for cases where the guided hole is needed (based on the pilot drilling) such that a range of BMD can be established instead of a single BMD value.

ANSWER – To clarify this point, a new equation (3) was introduced.

  • Figure 4 indicates Z-scores for different Vertebra regions. It is recommended to add information on Z-scores for bone density prior to figure 4.

ANSWER – the authors thank the reviewer. We removed the Z-score from the figure since is not the major subject of the discussion.

Reviewer 2 Report

This paper presented a new biomechanism (BKS) for screws and bone implants using a bone dental implant screw. In this biomechanism, the bone particles, blood, cells, and protein molecules removed during bone drilling are used as a homogeneous autogenous transplant in the same implant site, aiming to obtain primary and secondary bone stability, simplifying the surgical procedure, and improving the healing process. The subject is very interesting and can add knowledge to the metallic biomaterials field, mainly in dental implants. The introduction presents the problem to be studied, with current and adequate references. The methodology is usual for this kind of material. The results are very interesting and promising. The discussion of the results is robust and based on the earlier findings. The conclusions are based on the obtained results and presented discussion, showing that the bone inside the new model is denser at all four different synthetic bone densities. With the new biomechanical screw, it is possible to directly access the bone mineral density (BMD) in cases where more reliable data are needed during surgery treatments in patients with osteopenia to help confirm the diagnosis and treatment. I think the paper is adequate and can be accepted in its present form.

Author Response

About the Reviewer’s comments, we are answering their concerns in the following lines.

The authors would like again to acknowledge the Reviewer for their work on reading and suggesting improvements to the manuscript. We have addressed all the comments and the changes needed. Changes to the article were identified in blue.

We hope that the revisions in the manuscript and our accompanying answers will be sufficient to make our manuscript suitable for publication in the Mathematical and Computational Applications.

This paper presented a new biomechanism (BKS) for screws and bone implants using a bone dental implant screw. In this biomechanism, the bone particles, blood, cells, and protein molecules removed during bone drilling are used as a homogeneous autogenous transplant in the same implant site, aiming to obtain primary and secondary bone stability, simplifying the surgical procedure, and improving the healing process. The subject is very interesting and can add knowledge to the metallic biomaterials field, mainly in dental implants. The introduction presents the problem to be studied, with current and adequate references. The methodology is usual for this kind of material. The results are very interesting and promising. The discussion of the results is robust and based on the earlier findings. The conclusions are based on the obtained results and presented discussion, showing that the bone inside the new model is denser at all four different synthetic bone densities. With the new biomechanical screw, it is possible to directly access the bone mineral density (BMD) in cases where more reliable data are needed during surgery treatments in patients with osteopenia to help confirm the diagnosis and treatment. I think the paper is adequate and can be accepted in its present form.

ANSWER – The authors thank the reviewer for the careful reading of the manuscript, for the time spend, and for the constructive and positive comments.

Reviewer 3 Report

Manuscript ID: mca-1798891

Title: Increased Bone Density within a New Biomechanism

Special Issue: Numerical and Symbolic Computation: Developments and Applications 2021

https://www.mdpi.com/journal/mca/special_issues/SYMCOMP2021

The article was intended to present a new design of implant screw based on laboratory model without any biological or clinical background.

The authors of the article use the lofty expressions like ‘A new biomechanism’ (I  was not able to find out any in this text) and that implants were ‘created’ (screws are produced or designed), ‘new biomechanical screw’ (all implants are biomechanical screws!) and ‘This is ongoing research and further results, in the laboratory and in vivo, will confirm this study’ – maybe but probably will not.

They even claim a new stage of bone healing between primary and final stability which seems to be a myth at least on the base of this article.

Hence, before the new era in implantology claimed by the author of this article let’s face the facts – the current literature point outs that putting too much pressure on bone during implantation in aim to gain more primary stability may lead to more bone resorption during bone remodeling. By the way the implants with the holes are nothing new in implantology.

I think it is pointless to further mention the shortcomings of this study, which anyway cannot be considered for publication since the very similar texts have already been published at least twice:

1.       Andreucci, C.A.; Alshaya, A.; Fonseca, E.M.M.; Jorge, R.N. Proposal for a New Bioactive Kinetic Screw in an Implant, Using a Numerical Model. Appl. Sci. 2022, 12, 779. https://doi.org/10.3390/app12020779.

2.       Andreucci, C.A.; Fonseca, E.M.M.; Jorge, R.N. Advances and Current Trends in Biomechanics, 1st ed.; Taylor & Francis: London, UK, 2021; pp. 1–4.

Author Response

About the Reviewer’s comments, we are answering their concerns in the following lines.

The authors would like again to acknowledge the Reviewer for their work on reading and suggesting improvements to the manuscript. We have addressed all the comments and the changes needed. Changes to the article were identified in blue.

We hope that the revisions in the manuscript and our accompanying answers will be sufficient to make our manuscript suitable for publication in the Mathematical and Computational Applications.

The article was intended to present a new design of implant screw based on laboratory model without any biological or clinical background.

The authors of the article use the lofty expressions like ‘A new biomechanism’ (I  was not able to find out any in this text) and that implants were ‘created’ (screws are produced or designed), ‘new biomechanical screw’ (all implants are biomechanical screws!) and ‘This is ongoing research and further results, in the laboratory and in vivo, will confirm this study’ – maybe but probably will not.

They even claim a new stage of bone healing between primary and final stability which seems to be a myth at least on the base of this article.

Hence, before the new era in implantology claimed by the author of this article let’s face the facts – the current literature point outs that putting too much pressure on bone during implantation in aim to gain more primary stability may lead to more bone resorption during bone remodeling. By the way the implants with the holes are nothing new in implantology.

I think it is pointless to further mention the shortcomings of this study, which anyway cannot be considered for publication since the very similar texts have already been published at least twice:

  1. Andreucci, C.A.; Alshaya, A.; Fonseca, E.M.M.; Jorge, R.N. Proposal for a New Bioactive Kinetic Screw in an Implant, Using a Numerical Model. Appl. Sci. 2022, 12, 779. https://doi.org/10.3390/app12020779.
  2. Andreucci, C.A.; Fonseca, E.M.M.; Jorge, R.N. Advances and Current Trends in Biomechanics, 1st ed.; Taylor & Francis: London, UK, 2021; pp. 1–4.

ANSWER – The authors thank the reviewer for the careful reading of the manuscript and constructive comments. About our manuscript, this is ongoing research, and this study will guide the practical actions and tests.

The new Biomechanism it is already patented, and it is the modification of the flutes of a drill bit adapted in a screw for bone implants. The new biomechanism is a compactor of material (3.45 times denser as shown in this article) inside in through the screw.

The new stage will be the natural healing process that will occur inside of the screw. Since we have bone to bone contact, not only bone to implant contact (BIC) we will study the possibility of having fully bone anchorage for the screw. It will be confirmed or not in vivo on this ongoing research.

There is no more pressure made by the New biomechanism related to the screw into the bone. Because is also a drill bit that cuts the bone, in all the region of the threads of the screw, we will have a standardize pressure that will input less pressure in the bone then any kind of protocol applied today. The increased pressure is made by the bone into the bone.

The hole is not new, and therefore we adopted with safety in the new biomechanism. But cutting and collecting/compacting the bone inside and through the hole are the new biomechanical concept.

According our two previous publications are not about the new Biomechanism only. They are about the concepts behind it. Every publication shows different concepts that will guide the execution of Lab tests.

We hope that the referred answers were convenient clarified to make our manuscript suitable for publication.

Reviewer 4 Report

Introduction: Please, include some reference to how bone density aroun the screw improves its grip.

"In this ongoing research......the main objective is to simulate and test in vitro and in vivo......The new biomechanism will be mathematically described, ...... and have a protocol to be validated". Wouldn't it have been better to propose the publication of this work when all that was done? Or it is already done which allows the authors to conclude that "further results will confirm the results in laboratory and in vivo"?    

Delete "This could become the true definition of Ossointegration". It sounds excessive.

Materials and methods:  It is unclear if the screw has been tested in bone fresh or dry.

Why to test the screw in four different synthetic PCF foams? Please, explain the reason of that.

Conclusions: Are you  sure that "further results will confirm the results in laboratory and in vivo".

Author Response

About the Reviewer’s comments, we are answering their concerns in the following lines.

The authors would like again to acknowledge the Reviewer for their work on reading and suggesting improvements to the manuscript. We have addressed all the comments and the changes needed. Changes to the article were identified in blue.

We hope that the new revisions in our manuscript will be sufficient to make our manuscript suitable for publication in the Mathematical and Computational Applications.

Introduction: Please, include some reference to how bone density around the screw improves its grip.

ANSWER – The authors thank the reviewer. We changed in the introduction, including reference 5.

"In this ongoing research......the main objective is to simulate and test in vitro and in vivo......The new biomechanism will be mathematically described, ...... and have a protocol to be validated". Wouldn't it have been better to propose the publication of this work when all that was done? Or it is already done which allows the authors to conclude that "further results will confirm the results in laboratory and in vivo"?

ANSWER – Every step of the new biomechanical design is being tested before the experimental application. We believe that the mathematical concept of compacting material (Bone) inside described in this article it’s a robust proof of concept. The results in the lab and in vivo will be evaluated for dental implants using this approach as standard model; also, other researchers can use it in different fields of study, not necessarily dental implants.

Delete "This could become the true definition of Ossointegration". It sounds excessive.

ANSWER – Thanks for the suggestion. We adjusted the terms more accordingly.

Materials and methods: It is unclear if the screw has been tested in bone fresh or dry.

ANSWER – The screw was tested mathematically, numerically comparing the average densities of Human bone, and four synthetic bones with 4 classified densities for dental implants, hard to soft bone (1-4) with the volume inside the BKS and the compacting factor 3.45.

Why to test the screw in four different synthetic PCF foams? Please, explain the reason of that.

ANSWER – Thanks for the suggestion. We changed in the materials and methods introducing the reason.

Conclusions: Are you sure that "further results will confirm the results in laboratory and in vivo".

ANSWER – Thanks for the suggestion. We changed in the conclusions.

Round 2

Reviewer 3 Report

I have red the improved version of the manuscript- the problem is it can not be improved. I think it is pointless to mention the shortcomings of this study - it anyway cannot be considered for publication since the very similar texts have already been published at least twice.

Author Response

The authors would like to complement the opinion from the reviewer. About our manuscript, this is ongoing research, and this study will guide the practical actions and tests. Previous articles published by the authors deal with the same biomechanism, but for different studies, related to the state of the art and on a numerical model developed using the finite element method to study the mechanical behavior of the screw-bone under a dynamic load.

Reviewer 4 Report

You must also change the last phrase og the abstract in the same sense as the last phrase of conclusions.

Author Response

The authors would like again to acknowledge the Reviewer for their work on reading again our manuscript. We have addressed the changes needed in the abstract, identified in blue.

Round 3

Reviewer 4 Report

O.K. with your corrections but I think it will be better to published the work once completely finished.